task-sharing; non-specialists; mental health; health systems; training and supervision

**Corresponding author:**
Manaswi Sangraula;
Email: ms4983@cumc.columbia.edu

# The impact of task-sharing scalable mental health interventions on non-specialist providers: a scoping review

Manaswi Sangraula[1] , Josheka Chauhan[2], Chynere Best[3], Caroline McEneaney[2], Cheenar Shah[2,3], Adam D. Brown[2,4] and Brandon A. Kohrt[3]

[1]Heilbrunn Department of Population and Family Health, Columbia University Mailman School of Public Health, New York, NY, USA; [2]Department of Psychology, New School of Social Research, New York, NY, USA; [3]Center for Global Mental Health Equity, Department of Psychiatry and Behavioral Health, George Washington University, Washington, DC, USA and [4]Department of Psychiatry, New York University School of Medicine, New York, NY, USA

## Abstract

Task-sharing approaches that train non-specialist providers (NSPs), people without specialized clinical training, are increasingly utilized to address the global mental health treatment gap. This review consolidates findings from peer reviewed articles on the impact of task-sharing mental health interventions on NSPs at the individual, family and community level. Studies that highlighted facilitators, barriers and recommendations for improving the experiences of NSPs were also included in the review. Fifteen studies, conducted across eight countries, met the inclusion criteria. Seven studies were conducted in Sub-Saharan Africa, six in South and Southeast Asia and two studies were conducted in high-income countries in Europe. Benefits for NSPs included personal application of mental health skills, elevated community status and increased social networks. Challenges include burnout, lack of career progression and difficult workplace environments. Findings indicate that while there were many positive impacts associated with NSPs' work, challenges need to be addressed. Safety and harassment issues reported by female NSPs are especially urgent. Supervision, certifications, increased salaries and job stability were also recognized as significant opportunities. We recommend future intervention studies to collect data on the impact of intervention delivery on NSPs. Research is also needed on the impact of various supervision and health systems strategies on NSPs.

## Impact statement

While task-sharing approaches are increasingly used to address gaps in mental health care in low- and middle-income countries and high-income countries, evidence on how these interventions impact non-specialist providers (NSPs) is lacking. NSPs are individuals who have not previously received specialized mental health or clinical training. Though there is a growing body of literature that examines the experiences of NSPs, the majority of studies and trials focus on participants' outcomes without examining the personal-, familial- and community-level impacts on NSPs. To our knowledge, this is the first review that consolidates current findings on benefits and challenges of these interventions on NSPs, as well as the training/supervision, organizational- and systems-level facilitators, barriers and recommendations to increase the positive impacts and decrease the negative impacts of task-sharing interventions on NSPs. Many of the recommendations made by NSPs and included in the review are related to how task-sharing interventions can strengthen training and supervision and link NSPs to existing health systems not only to better implement the intervention but also to develop a career pathway for NSPs. Therefore, this review highlights facilitators and recommendations for researchers and implementers working to increase the sustainability of task-sharing interventions.





## Introduction

Shortages in health-care professionals, especially for mental health, are issues faced by many health systems in low- and middle-income countries (LMICs) (Bruckner et al., 2011). Task-sharing refers to working with non-specialists providers (NSPs) to deliver health care traditionally provided by people with professional degrees (World Health Organization [WHO], 2007). Within the context of global mental health, NSPs are individuals who provide mental health care but have not previously received specialized mental health or clinical training (Raviola et al., 2019; Van Ginneken et al., 2013). NSPs may also be referred to as community mental health workers, lay counselors, lay providers or other program specific titles (Bunn et al., 2021) and are often trusted

figures within the community (Chibanda et al., 2011). Task-sharing has increasingly been promoted by the WHO as a crucial approach for increasing access to mental health care and for reducing health disparities (WHO, 2020b, 2021a). As a form of task-sharing, NSPs can be trained to deliver brief mental health interventions (Patel et al., 2018). Numerous studies have demonstrated that these interventions are effective in reducing symptoms of depression, anxiety and increasing daily functioning (Barbui et al., 2020; Naslund & Karyotaki, 2021). In recent years, these interventions have also been delivered in high-income countries (HICs) especially to provide care for populations traditionally marginalized by health-care systems (Giusto et al., 2024; Turan et al., 2023).

However, despite the strong justification for utilizing NSPs to deliver task-sharing interventions, important questions remain on how and to what extent they are personally impacted by participating in the training, supervision and delivery of these interventions. Research has revealed challenges faced by NSPs, such as lack of motivation due to stress and burnout (Strachan et al., 2015). These challenges may not only reduce the quality and effectiveness of intervention delivery but are also alarming from a well-being perspective (Wall et al., 2020). Though NSPs may be exposed to the same daily social, economic and political stressors as their clients (Verhey et al., 2021), they are often regarded as delivery agents who can "help close the mental health treatment gap" rather than persons with unique skills and perspectives that may also need additional support (Maes, 2015; Maes et al., 2018).

Although the majority of mental health intervention trials focus on participant outcomes, there is a growing body of literature that examines the experiences of NSPs in delivering task-sharing interventions and the impact of their involvement on themselves as well as their families and communities. However, there has not yet been a systematic or scoping review summarizing the evidence on the impact of these interventions on NSPs. This scoping review aims to consolidate research on how training, supervision and delivery of mental health interventions impact NSPs at the individual, family and community level. The review also aims to summarize existing recommendations at the programmatic and organizational levels to increase the positive impact and mitigate any negative impacts of these interventions on NSPs.

This review was guided by two questions: 1) What are the individual-, family- and community-level impacts of task-sharing mental health interventions on NSPs? and 2) What are individual-, programmatic- and organizational-level recommendations to increase the positive impact of task-sharing mental health interventions on NSPs? This scoping review aims to map and summarize findings on this little researched topic, report proposed recommendations and identify additional gaps in literature. These criteria were categorized according to the Population, Concept and Context framework (Briggs, 2014) (Table 1).

## Methods

### Protocol and methodology

Our scoping review protocol was developed using the five-stage methodological framework proposed by Arksey and O'Malley (2005). This included: 1) identifying the research question, 2) identifying relevant studies, 3) study selection, 4) charting the data, and 5) collating, summarizing and reporting the results (Arksey & O'Malley, 2005). The final protocol was registered with the Open Science Framework on 9 April 2024 (osf.io/52b3z). The methodology

**Table 1.** Inclusion criteria

| Inclusion criteria | Definition |
| --- | --- |
| Population | Non-specialist providers (NSPs)<br>• The World Health Organization's (WHO) definition of a community health workers or lay providers are persons who have received some (mental) health training (up to 2 years) but are not considered mental health specialists (World Health Organization, 2020a). They live or have close proximity to the communities that they work with.<br>• May be referred to as: Community Health Workers (CHWs), Lay Health Workers (LHWs), lay providers, lay mental health workers (LMHWs), Counselors and other terms. These terms will be used interchangeably throughout this review.<br>• Studies focused on "peer specialists" or providers who had specific lived experience of mental health conditions were excluded from the scoping review.<br>• Studies that only included NSPs in their sample to report on participant or program related outcomes, such as perceived effectiveness, acceptability and reach of the intervention, were excluded from the study.<br>• Only studies that reported on the experiences or outcomes of NSPs that were involved in scalable mental health task-sharing interventions were included in the search. |
| Concept | Scalable mental health interventions<br>• A brief and defined mental health intervention for people experiencing mild to moderate mental health conditions.<br>• Studies focused on NSPs providing general mental health care rather than a specific task-sharing mental health intervention were excluded from the review.<br>• Scalable mental health interventions for all age groups and populations, including children, adolescents, adult women and men, were included in the review.<br>Impact<br>• Individual, family and community level<br>Recommendations<br>• Individual-, programmatic- and organizational-level recommendations to increase the positive impact of task-sharing scalable mental health interventions on NSPs. |
| Context | Low- and middle-income countries (LMICs), high income countries (HICs), Urban, Rural |

is presented according to the PRISMA-ScR guidelines (Tricco et al., 2018).

### Data sources and search strategy

Two researchers (MS and JC) developed a comprehensive plan for the scoping review, including the identification of the search terms and search procedures. We systematically searched PubMed, NLM, Biomedcentral, ScienceDirect, MEDLINE and PsychInfo. The search included terms such as "task-sharing mental health intervention," "community health workers," "lay health workers," "nonspecialists," "scalable mental health interventions," "task sharing interventions," "perceived impacts," "nonspecialist training and supervision," "LMICs," "mental health conditions" and "facilitators and barriers." These terms were combined using "OR" and "AND" operators (please see Supplementary Material for complete search strategies). We also scanned the references section of included articles

to identify additional relevant studies. The search was conducted between May and July 2023. All search results were imported into an excel document to manage references more efficiently.

### Eligibility criteria

We searched for peer-reviewed articles that were published in English-language journals over the last 20 years (2003–2023) and met the following criteria and definitions: 1) presented quantitative and/or qualitative findings on the individual-, family- or community-level impacts of task-sharing interventions on NSPs, and 2) highlighted facilitators, barriers and recommendations on increasing the positive impact of task-sharing interventions on NSPs, specifically characteristics of NSPs who would be successful at task-sharing interventions, recommendations for their training and supervision and organizational-level recommendations. Studies

conducted in both LMICs and HICs were eligible for the scoping review. HICs were also included in the search because HICs are increasingly using task-sharing models, especially to increase access to mental health care within minoritized communities (Giusto et al., 2024). We excluded studies that did not include NSPs as participants or within the study sample.

### Data extraction

We maintained a structured sheet for detailed extraction and high-lighted key findings across the studies according to the main themes, such as mental health and social impacts at the individual, family and community level, and facilitators, barriers and recommendations on increasing positive impacts of task-sharing interventions on NSPs. This approach allowed for a systematic and comprehensive extraction of data, ensuring that both overarching themes and

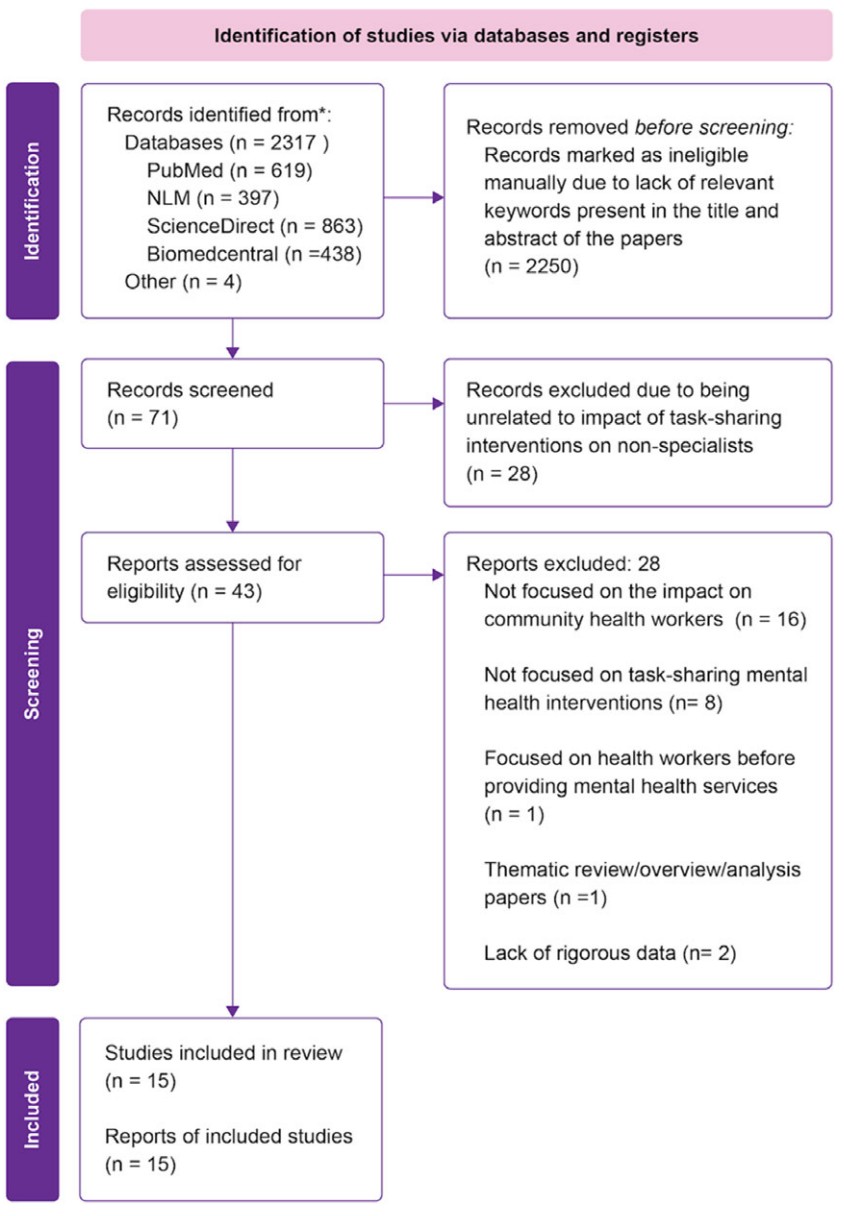

**Figure 1.** Identification of studies flow diagram.

specific details from each paper were captured for a nuanced understanding of the literature. Our search and data extraction process were conducted by the first and second author, who completed a double review at all stages of the screening and met weekly during the search, extraction and analysis process to share feedback and ensure quality. Data in the extraction sheet was summarized by theme and is shared in the results section.

## Results

### Literature search

Initially, a total of 2,317 records were identified through searching databases (Figure 1). After screening by title and abstract, 71 articles were relevant to the defined research questions and met all our inclusion criteria. These 71 records were screened using their full text to determine if they fit the inclusion criteria. After excluding 28 articles, a total of 43 records were reviewed by the first two authors by doing an in-depth reading of the full text and ensuring that it met the eligibility criteria. Thirty-two records were excluded. An additional four records were found by scanning the references section of the included articles. Fifteen articles were included in the review. Most articles were excluded because they did not collect data on the impact of task-sharing interventions on NSPs or recommendations for improving their outcomes, but instead focused on NSPs' perspectives on feasibility and acceptability of mental health interventions. Several articles were also excluded because they provided a general overview on the impact on NSPs rather than collecting specific quantitative or qualitative data.

### Study characteristics

Of the 15 included studies, 7 of the studies were conducted in Sub-Saharan Africa (2 in Kenya, 2 in Zimbabwe and 3 in South Africa), 6 in South and Southeast Asia (3 in India, 2 in Pakistan and 1 in Vietnam) and 2 of the studies were conducted in HICs in Europe (one in the Netherlands and one in Switzerland) (Table 2). Some NSPs from the two studies in HICs were refugees and migrants themselves and were delivering mental health interventions to refugee and migrant communities. Regarding data collection, 12 studies gathered qualitative data from NSPs, 1 study conducted quantitative assessments and 2 studies used both approaches. In terms of recommendations and impacts, two studies discussed impact on the participants while two made recommendations for improving non-specialist outcomes. Fourteen studies gathered data on both the impact and recommendations. NSPs were referred to by various titles in the studies including lay health workers, counselors, helpers, peer volunteers, female volunteer community health workers and social collaborators. For reporting findings in this review, all the providers will be referred to as NSPs.

### Individual-, family- and community-level impacts on non-specialist providers (NSPs)

### Individual-level mental health and social impacts

NSPs in eight studies reported that intervention training, delivery and supervision led to learning skills that bettered their mental health (Figure 2). They applied intervention skills, such as problem solving, to their own lives (Atif et al., 2019). They improved their mental health literacy, coped better with their personal day-to-day stressors, and these personal benefits were useful in helping them stay motivated in delivering the intervention (Abas et al., 2016; Jacobs et al., 2021; Wall et al., 2020; Woodward et al., 2022). NSPs in HICs noted how their clients' stories affected them. For example, NSPs in the Netherlands described that participants sometimes shared "intense" stories that personally impacted them (Woodward et al., 2022). NSPs in Switzerland noted that they could relate to their clients' sadness and over time became "more resilient and less likely to be negatively affected by people's stories" (Spaaij et al., 2023). Because NSPs were providing services to fellow refugees and migrants in the Netherlands, they mentioned feeling motivated to provide support knowing that it would be difficult to reach a psychologist in the context of their culture (Woodward et al., 2022). NSPs also noted that over time they were able to be more empathetic (Giusto et al., 2021; Munodawafa et al., 2017). Additional personal benefits included gaining a sense of control over their own lives, increased self-confidence, increased interpersonal skills, sense of self and ability to prioritize self-care (Atif et al., 2016; Wood et al., 2021).

A common theme across seven of the studies was that the NSPs experienced a change in their relationship with their communities. Some NSPs reported that their community status was elevated because of their involvement in task-sharing programs, as they became increasingly recognized for their skills and expertise (Atif et al., 2019; Verhey et al., 2021; Wood et al., 2021). NSPs in Vietnam expressed pride in their role as community members increasingly turned to them for consultation and support (Chau et al., 2021). NSPs in Kenya reported being seen as more "approachable" by other community members and strengthening bonds with local institutions, such as churches and schools (Wall et al., 2020). In India, NSPs expressed that their involvement in the training, supervision and delivery of the interventions challenged stereotypical gender norms and led to a sense of empowerment by "not caring what others think and doing the training for themselves" (Abas et al., 2016). They saw themselves as "agents of change" in their communities and experienced satisfaction and personal reward from their new role (Abas et al., 2016). Positive endorsement for their role by their own communities and families contributed to their motivation to deliver task-sharing interventions (Atif et al., 2016).

Delivering task-sharing interventions increased social networks and the potential for future community facing opportunities, especially for women who previously had little opportunity to socialize outside of their families in conservative and rural settings (Atif et al., 2019). Taking part in a training and receiving a certificate was seen as a pathway toward future opportunities (Dev et al., 2022). NSPs reported gaining trust, praise and forming good relationships with neighboring families (Atif et al., 2019; Dev et al., 2022). They expanded their social networks by forming friendships and support systems with other peer NSPs (Atif et al., 2016). NSPs in Pakistan reported gaining new skills and confidence, which increased opportunities for employment and upward mobility. NSPs found jobs as teaching assistants, community health workers and elected village councilors to represent other local women, which they attribute to the expansion of their social networks (Atif et al., 2019).

> "I got elected as a Lady Councillor to represent my village in the community. This was possible because I met many families working as a volunteer. Was able to help them through difficult situations and this gave me the confidence to extend my role. I am very happy with my new role." – NSP, Pakistan

> (Atif et al., 2019)

**Table 2.** Characteristics of included studies

| Author(s) (year) | Study design | Location | Intervention setting | Provider description | Target population | Task-sharing intervention | Training and supervision | Assessments utilized | Review aspect |
|---|---|---|---|---|---|---|---|---|---|
| Abas et al. (2016) | Friendship Bench Case Study | Zimbabwe | City of Harare Health Education Unit | Female lay health workers | Patients with depression and other CMD | Friendship Bench Program | Friendship Bench Intervention (Pre-Trained) | Quantitative and qualitative, 4–8 years post study, 5 focus groups | Impact on LHW families Individual-level facilitators, barriers, recommendations (f/b/r) Program-level f/b/r Organizational-level f/b/r |
| Atif et al. (2016) | THPP Pilot Interviews | Rawalpindi, Pakistan | Rural sub-district of Rawalpindi, Punjab, Pakistan | Peer volunteers (PVs) | Mothers experiencing perinatal depression | Thinking Healthy Program Pakistan (THPP) | PVs Training (Pilot Phase of THPP) | Qualitative interviews between 2013 and 2014, (n = 8) | Mental health/social impact Individual-level f/b/r |
| Atif et al. (2019) | Mixed Methods | Rawalpindi, Pakistan | Clinics | Female peer volunteers | Women with perinatal depression | THPP (Thinking Healthy Program Peers) | Trained and supervised by non-specialist THPP facilitators | Quantitative and qualitative over a 5-year period (n = 31) | Mental health/social impacts Individual-level f/b/r Organizational-level f/b/r |
| Chau et al. (2021) | Semi-structured interviews | Vietnam | Local health centers | Social collaborators/village health workers | Community members experiencing social difficulties | Supportive Self-Management Task-Sharing Intervention (SSM) | MAC-FI study social collaborators training | Qualitative descriptive methods (n = 47) | Impact on LHW Program-level f/b/r Organizational--level f/b/r |
| Dev et al. (2022) | PRIME Post-Programme Interviews | India | Local Health Care Centers | ASHAs (female volunteer CHWs) | Participants at local health care centers | PRIME ASHAs | Public mental health training for community screening, first aid, referral and follow-up | Qualitative interviews post-PRIME, 2014–2016 (n = 12) | Mental health/social impact, Organizational-level f/b/r |
| Giusto et al. (2021) | Training Program Evaluation (Mixed-Methods) | Kenya | Rift Valley Province of Kenya in a peri-urban community surrounding the town of Eldoret | Peer-fathers, with no previous mental health training | Clients who were recruited through community referrals | LEAD (Learn, Engage, Act, Dedicate) | Nominated fathers lay-counselor training (10-day program) | Qualitative and quantitative (n = 11) | Mental health/social impact Impact on LHW Impact on LHW families Program-level f/b/r |
| Jacobs et al. (2021) | In-depth Interviews: MIND Project | South Africa | Primary care facilities participating in the project MIND cluster RCT | FBCs trained for HIV adherence counseling services | Chronic disease patients at-risk for a CMD | Motivational Interviewing and Problem-solving Therapy (MI-PST) | Blended MI-PST during cluster RCT of MIND Project | Qualitative interviews (n = 18) | Impact on LHW Individual-level f/b/r Program-level f/b/r Organizational-level f/b/r |
| Munodawafa et al. (2017) | AFFIRM-SA Counselors Interviews | Khayelitsha, South Africa | Clinics | Lay counselors | Clients with perinatal depression | Psycho-social interventions | CHWs trained/supervised by mental health counselors (MHC); additional support by CL | Qualitative (n = 6) | Impact on LHW Individual-level f/b/r Program-level f/b/r Organizational-level f/b/r |
| Pereira et al. (2011) | Cluster RCT: Primary Health Facilities | India | Primary health care facilities (PHCs) and private GP practices | Lay health counselor (LHC) | Patients with depression and other CMD | MANAS Collaborative Stepped Care Intervention | LHCs pre-trained in MANAS intervention | Qualitative (n = not specifically mentioned for LHCs) | Individual-level f/b/r Program-level f/b/r Organizational level |

**Table 2.** (*Continued*)

| Author(s) (year) | Study design | Location | Intervention setting | Provider description | Target population | Task-sharing intervention | Training and supervision | Assessments utilized | Review aspect |
|---|---|---|---|---|---|---|---|---|---|
| Spaaij et al. (2023) | PM+ Implementation Interviews | Switzerland | Outpatient Clinic for Victims of Torture and War of the University Hospital Zurich | Problem management plus (PM+) helpers | Refugees and asylum seekers | PM+ | Helpers pre-trained in PM+ (STRENGTHS Project) | Qualitative (*n* = 5) | Mental health/social impact Individual-level f/b/r Program-level f/b/r |
| Verhey et al. (2021) | Survey: Friendship Bench LHWs | Zimbabwe | 70 clinics that adopted the Friendship Bench (FB) program | Lay health workers (LHWs) with FB training between 2013 and 2016 | Patients with depression and other CMD | Friendship Bench Program | Friendship bench intervention (pre-trained) | Quantitative (*n* = 182) | Mental health/social impact Impact on LHW Impact on LHW families Program-level f/b/r Organizational-level f/b/r |
| Wood et al. (2021) | Semi-structured interviews | India | 5 non-governmental organizations across India | Lay mental health workers in NGOs | Vulnerable populations | Basic counseling skills | Basic Counseling Skills Training | Qualitative and quantitative (*n* = 32) | Mental health/social impact Impact on LHW Impact on LHW families Individual-level f/b/r Program-level f/b/r Organizational-level f/b/r |
| Woodward et al. (2022) | RCT Stakeholder Interviews | Netherlands | Dutch mental health and psychosocial support (MHPSS) system | PM+ helpers | Refugees participating in the STRENGTHS' RCTs | Syrian REfuGees MeNTal HealTH care systems (STRENGTHS) | Syrian Refugees Training in PM+ (STRENGTHS RCTs) | Qualitative (*n* = 20) | Mental health/social impact Impact on LHW Organizational-level f/b/r |
| Wall et al. (2020) | Lay Counselor Interviews | Kenya | Peri-urban communities surrounding the town of Eldoret, Kenya. | Lay counselors | Families experiencing difficulties in their relationships and family dynamics | Tuko Pamoja (TP)-family therapy intervention | TP Counselors Categorized by Experience (Moderate, Minimal, Training Only) | Qualitative (*n* = 20) | Mental health/social impact Organizational-level f/b/r |
| van de Water et al. (2017) | Nested Qualitative RCT Stakeholder Study | Stellenbosch (SA) | Stellenbosch University, South Africa | Nurse counselors | Adolescents (ages of 13–18 years old) with PTSD | Prolonged exposure therapy for adolescents (PE-A), supportive counseling (SC) | Registered Nurses Training in PE-A and SC (Advanced Psychiatry Diploma) | Qualitative (n = 6) | Mental health/social impact Program-level f/b/r Organizational-level f/b/r |

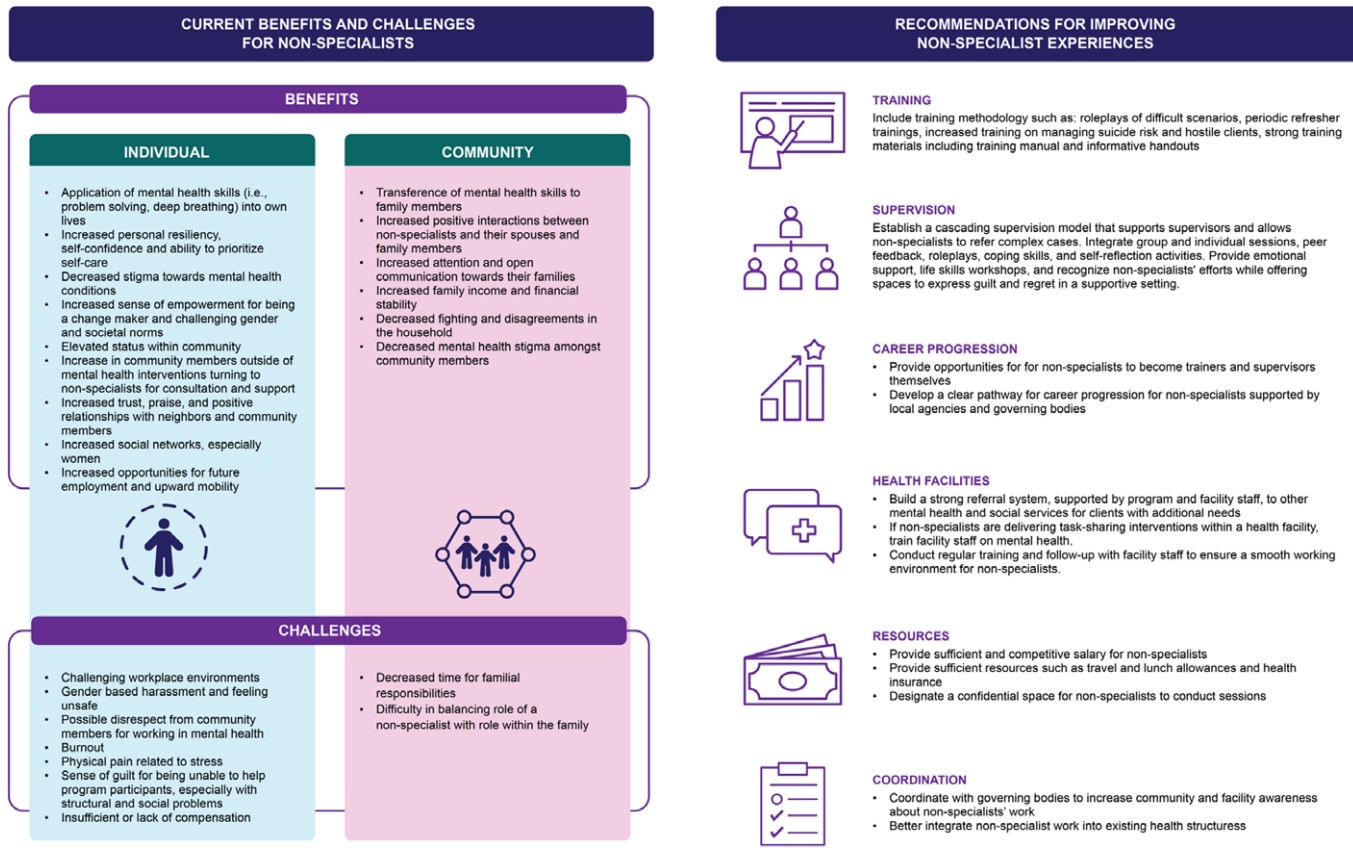

**Figure 2.** Impact of task-sharing interventions on non-specialist providers (NSPs) and recommendations for improving non-specialists' experiences delivering interventions.

Seven studies described individual-level negative consequences of taking part in task-sharing interventions. NSPs in India shared that their work often lacked flexibility and they were sometimes asked to work long hours, over the weekend or until the evening (Wood et al., 2021). NSPs cited their physical environment as a challenge and described traveling to unfamiliar areas as "the most negative thing" and a "nightmare" (Van de Water et al., 2017). Female NSPs shared experiences of gender-based harassment and feeling unsafe, especially when traveling in the dark, conducting sessions in areas with high crime and doing home visits with male clients (Munodawafa et al., 2017; Wood et al., 2021). NSPs in India shared scenarios in which they were disrespected by community members for implying that their family members have mental health problems or intruding in personal matters (Dev et al., 2022). NSPs in India also noted that they faced stigma from family and community members for working with people of lower caste (Wood et al., 2021). NSPs in Kenya mentioned experiences of burnout or feeling "so compressed or stretched [by their work] they felt they could not continue working" (Wall et al., 2020). They expressed physical symptoms, such as pain, fatigue and feeling sick, when they were burned out and overwhelmed (Wall et al., 2020). The overwhelming nature of the work led NSPs in the Netherlands to drop out leaving a greater workload for the remaining NSPs (Woodward et al., 2022).

NSPs in South Africa also shared their struggles knowing that outside of the task-sharing program, participants would be returning to "dysfunctional systems" and felt a sense of guilt for not being able to do more (Van de Water et al., 2017). Structural and social difficulties, such as problems with daily wages, seasonal work and alcoholism, often felt beyond the NSPs' influence and they felt guilty for not being able to refer patients to appropriate services, which unfortunately did not exist in their communities (Abas et al., 2016; Munodawafa et al., 2017; Pereira et al., 2011; Van de Water et al., 2017). Because they knew that participants lacked access to other functional services, NSPs approached their work with "grave responsibility" and shared difficulties in keeping a professional distance and navigating boundaries (Van de Water et al., 2017).

> "When I started there I, yoh, took everything home with me … Until the school psychologist told me: you cut off! Because one night I was in such a state about a child that I started to shake. And my husband said: you can't carry on like this. You must cut off! Then I said: but how do you sleep if you know that the child might not be safe? …" – NSP, South Africa (*Van de Water* et al., *2017*)

Wall et al. (2020) reported that a key component of this burnout experience was a mismatch between NSPs' genuine desire to help and their clients sometimes not engaging or making progress. This led to NSPs to question their counseling capabilities and feel "burdened in my heart" (Wall et al., 2020). NSPs also felt a sense of helplessness (Dev et al., 2022). However, seeing positive changes in their clients led NSPs to feel proud of their accomplishments (Dev et al., 2022; Giusto et al., 2021; Spaaij et al., 2023; Wall et al., 2020) and even reduced feelings of burnout (Wall et al., 2020).

Only two studies explored rates of distress among NSPs using quantitative methods. Post-traumatic stress disorder and common mental health disorder were identified among 6% and 11% of 204 NSPs in Zimbabwe (Verhey et al., 2021). Almost half of the LHWs were widowed (49%) and 79% relied on their own salary for income. Many reported suffering from a chronic illness (70%) and 9.3% reported currently living with HIV. Several NSPs reported negative life events within the last 6 months including death of a

family member, serious illness, loss of accommodation, domestic violence and assault. Wall et al., 2020 used Participatory Risk Mapping to identify that poverty, family issues, providing counseling and balancing their roles were categorized as top issues among NSPs. Authors highlighted that though NSPs are tasked with the heavy lift of reducing the mental health treatment gap, they often face similar daily stressors and simultaneously are exposed to vicarious trauma through their clients (Verhey et al., 2021).

### Family- and community-level mental health and social impacts

Three studies highlighted that the skills NSPs learned also impacted their family members. This was seen as a personal benefit and helped NSPs stay motivated (Abas et al., 2016). In Kenya, NSPs reported positive changes in how they interacted with their spouses and disciplined their children (Wall et al., 2020) (Figure 2). Male NSPs in Kenya mentioned that they practiced more open communication and increased attention to their families (Giusto et al., 2021). They also shared skills, such as time management, with their children and reported using problem-solving and communication skills, to control personal reactions to negative emotions, leading to positive changes in their families (Wall et al., 2020). NSPs reported decreased fighting, increased time spent together, and overall "increased love and togetherness" in their homes (Wall et al., 2020).

> "I can say that I used to be very harsh and judgmental, giving final answers to issues; unlike now I know where I have come to learn that counseling is a process … now I can sit with my wife and children as a family and I listen to them so that I can be able to help them, because as a parent, I am also a counselor at home." – NSP, Kenya
> (Wall et al., 2020)

One study noted that NSPs had an important impact on community mental health. Prior to receiving services from NSPs in Vietnam, families had previously attributed mental health conditions to sorcery and infections but NSPs helped to reduce stigma about mental illness within clients' families and the wider community (Chau et al., 2021). NSPs were able to make this impact through client home visits and building "good relationships" with community members.

Three studies mentioned the negative impacts that NSP work had on their own families. NSPs shared that the reality of "sacrificing time" to work left less time for familial responsibilities (Wall et al., 2020). NSPs in India mentioned that the expectation of working long hours, over the weekend or during evenings led them to miss spending quality time with their families (Wood et al., 2021). They also found it challenging to start a position in a field they were new to and to balance expectations of this role with the needs of their own families (Giusto et al., 2021; Wall et al., 2020). For a majority of NSPs (79%) in Verhey et al.'s (2019) study, a small monthly salary from delivering mental health care was their only source of income and because of the high unemployment rate in Zimbabwe, this income was a source of financial stability that supported their families as well. However, as the sole provider of their families, some NSPs had to balance their time delivering mental health interventions with other income generating activities, especially if they were not paid for their involvement (Wall et al., 2020).

### Facilitators, barriers and recommendations for increasing positive outcomes for NSPs

### Individual-level facilitators and barriers

Two studies identified several individual characteristics that facilitated success for NSPs and protected against burnout. NSPs shared

a general desire and motivation to better the mental health of others (Woodward et al., 2022). Some shared a "deep-rooted calling from God," especially after witnessing widespread problems in their communities (Wall et al., 2020). Ability to give and receive critical feedback, desire to challenge oneself and ability to persist in following up with clients were seen as personal qualities that were helpful in overcoming barriers in delivering care (Wood et al., 2021).

> "With work experience … there is development. There may be some challenge but, in the future, there will be a more difficult one, and for that, I have to prepare myself. And I will not step back. I will face [the challenge] and see if I can tackle [the challenge], I can surely tackle [the challenge]. This self-confidence helps me to move ahead." – NSP, India
> (Wood et al., 2021)

Ability to compartmentalize work and home life and strong communication skills were identified as protective skills against burnout (Wood et al., 2021). NSPs in India shared that some family members were opposed to them continuing their role but support from at least one family member, such as a spouse or child, helped NSPs face family resistance and provided them with practical and emotional support (Wood et al., 2021). Respect, encouragement and empowerment from fellow NSPs, supervisors, family and community members were helpful in sustaining motivation, especially when clients were not progressing (Wall et al., 2020).

### Training and supervision facilitators, barriers and recommendations

NSPs in three studies reported that supervision and peer support strengthened confidence and enhanced protective factors against burnout (Figure 2). Training methods, such as role-plays of difficult scenarios, helped NSPs gain confidence to overcome potential challenges (Pereira et al., 2011). Supervision made NSPs feel like they were part of a team and were receiving recognition for their hard work (Wall et al., 2020). Expressing feelings of guilt and regret to supervisors and peers were seen as stress relievers and supported in managing personal reactions and navigating professional boundaries (Wall et al., 2020). An NSP in South Africa mentioned that she learned to stop internalizing difficult stories that her clients shared with her and manage her own anxiety through breathing and other techniques during supervision (Van de Water et al., 2017). NSPs also found reassurance in supervision, knowing that adolescents would be referred appropriately and that supervisors would step in to support more difficult cases (Van de Water et al., 2017).

NSPs across five studies expressed various levels of satisfaction with training and supervision (Chau et al., 2021; Van de Water et al., 2017). They expressed wanting to feel better equipped to handle difficult problems and requested additional trainings and support to enhance their counseling skills (Wall et al., 2020). Because NSPs often had a range of backgrounds, prior occupations and limited experience with short-term trainings, strong training and supervision were necessary to assure that the trainees delivered care of similar quality and to avoid implementation barriers (Chau et al., 2021; Spaaij et al., 2023). NSPs also valued receiving support from peers during group supervision sessions (Van de Water et al., 2017; Verhey et al., 2021). However, an NSP in South Africa noted that while supervision was helpful, she found it difficult to receive supervision only in group-based settings because other peers were more extroverted than her (Van de Water et al., 2017). NSPs sometimes felt that they could not meet the expectations of their supervisors, which led to feeling hopeless and wanting to avoid supervision (Wall et al., 2020).

Seven studies recommended the integration of additional components in training and supervision including time to troubleshoot expected challenges before seeing clients, teaching coping skills for NSPs, opportunities for NSPs to share their responses to client interaction (Wall et al., 2020); supervision by more experienced social workers, refresher trainings and long-term supervision (Chau et al., 2021); structured "real-life" role-plays (Spaaij et al., 2023); additional instruction on building strong counselor–client relationships, tracking behavioral changes in clients (Giusto et al., 2021); non-specialist competency assessments (Spaaij et al., 2023); additional training managing suicide risk and interacting with hostile clients (Abas et al., 2016); structured opportunities for self-reflection and de-briefing (Jacobs et al., 2021; Munodawafa et al., 2017) and emotional support services to prevent and reduce secondary traumatization (Jacobs et al., 2021). NSPs in South Africa attended workshops that focused on trauma debriefing and managing personal finances (Munodawafa et al., 2017). These workshops assisted NSPs in coping with their roles and preventing them from feeling burned out or overwhelmed.

NSPs across six studies noted the importance of training materials and innovative supervision models. Training manuals and supervision handouts provided additional reassurance for NSPs that they were delivering care with fidelity (Munodawafa et al., 2017; Wall et al., 2020). Studies also emphasized the importance of collaborative peer support through formalized meetings or informal communication (e.g., WhatsApp groups) (Verhey et al., 2021; Wall et al., 2020). Supervisors also expressed a need for supervision and support for themselves (Abas et al., 2016; Munodawafa et al., 2017; Spaaij et al., 2023). To address this, a study in South Africa used a cascading supervision model where NSPs provided supervision to NSPs, which was monitored by a psychologist (Jacobs et al., 2021). It was noted that supervision by NSPs rather than psychologists may have reduced power differentials leading to improved supervision. If NSPs were to be trained as supervisors, helpers in Switzerland noted that they would need additional training on facilitating dialog, trainings and workshops, as well as planning and time management skills (Spaaij et al., 2023).

### Organization- and systemic-level facilitators, barriers and recommendations

NSPs across five studies expressed a need for additional compensation for their work (**FIGURE 2**). Altruism and personal satisfaction were not enough to sustain NSPs' motivation and involvement in task-sharing interventions (Abas et al., 2016; Woodward et al., 2022). Having sufficient resources, such as salary, travel allowance, lunch allowance and health insurance, was cited as a facilitator for NSPs to carry out their work (Wood et al., 2021). Abas et al. (2016) hypothesized that the lack of financial incentives for the NSPs may have impacted performance, such as lower-than-expected follow-up appointments and poor documentation. Though recruiting NSPs was challenging because of the difficult nature of the work, NGOs that paid competitive salaries were able to more easily recruit and retain NSPs (Wood et al., 2021). Remuneration was cited as a source of job satisfaction, motivation and empowerment (Verhey et al., 2021) and allowed many NSPs, especially women, to gain financial stability and independence outside of their families' influence (Dev et al., 2022).

> "… if any person is working, they work with the hope that they will gain something. […] Otherwise her family members put her down by asking why she is roaming around all day and when would she do

the household work, when would she take care of the kids. And so, if we get money, our families would not say anything." – NSP, India (Dev et al., 2022)

Five studies pointed to the need for a change in health systems to better integrate NSP work into existing health structures. NSPs in South Africa discussed the importance of educating facility staff in primary care and other governing bodies on the basics of mental health, the work that the NSPs are doing and the resources, such as adequate compensation, time and confidential space, that is needed for their success (Jacobs et al., 2021). NSPs reported feeling emotionally burdened when they lacked sufficient supplies and were not welcomed in certain venues such as schools and clinics to carry out their work (Munodawafa et al., 2017; Van de Water et al., 2017). Increasing mental health literacy among primary health-care workers may create a more welcoming climate for NSPs for task-sharing implementation (Jacobs et al., 2021). Strong referral systems (Munodawafa et al., 2017) and comprehensive primary care (Abas et al., 2016) provided additional institutional backing for NSPs. For example, NSPs suggested additional support from primary care centers to help them navigate systemic challenges, such as linking clients to additional services (Jacobs et al., 2021). A study noted that their NSPs were "inadequately compensated and overburdened with multiple responsibilities" and additional coordination with the Ministry of Health and other governing bodies is needed to strengthen policies related to NSPs' roles, training and compensation (Chau et al., 2021).

Two studies recommended establishing a clear pathway for progression and continued career development for NSPs. NSPs described that their overall morale and stress levels were negatively impacted by the lack of job security, especially since their work was often funded by time-limited grants (Atif et al., 2019; Wood et al., 2021). Atif et al., 2019 recommended for governmental and non-governmental agencies to adopt mental health programming to provide a sense of certainty for career progression after completion of a task-sharing study. For example, NSPs in Pakistan, who received strong community feedback and scored high on competency assessments, shadowed supervisors to become peer-supervisors. The authors stressed the importance of systemic approaches to evaluating competency that could lead NSPs to take part in scaled-up programming. NSPs in India also suggested for NGOs to provide a certification of skills gained in delivering task-sharing interventions that could help with future employment (Wood et al., 2021). Professional development opportunities and having a clear career path would incentivize NSPs as well as increase the sustainability of programming (Atif et al., 2019).

### Discussion

This scoping review synthesized the individual-, family- and community-level impacts of task-sharing interventions on NSPs. The review also included programmatic-, organizational- and system-level barriers, facilitators and recommendations for strengthening the positive impact of these interventions on NSPs. Findings from this scoping review revealed that NSPs face work-related stressors, such as high workload, challenging workplace environments and lack of role clarity, that are similar to mental health specialists (O'Connor et al., 2018). While there are numerous studies examining the impact of service delivery on mental health specialists (McCormack et al., 2018; Vivolo et al., 2024), this scoping review found that relatively few studies have examined the impacts of these interventions on the delivery agents themselves. To our knowledge,

this is the first scoping review to consolidate findings related to the impact of mental health interventions on NSPs involved in delivering the interventions. Because of the limited research on this topic, we recommend for more task-sharing intervention studies to collect qualitative and qualitative data on the impacts of mental health intervention delivery on NSPs.

While the NSP role was associated with many positive impacts, several studies from the scoping review highlight safety, harassment and discrimination as key stressors for female NSPs, who make up the majority of the global community health workforce (Perry et al., 2014). Safety of female NSPs, especially in conflict affected settings (Raven et al., 2015), is widely acknowledged as a concern in the global health workforce (Dasgupta et al., 2017; Fotso, 2015; Razee et al., 2012). However, a recent article poignantly noted that most studies evaluating community health programs mention experiences or threats of harassment and violence toward NSPs as a secondary finding (Closser et al., 2023) rather than calling for action. The lack of formalized and sustainable systems of protection by employers leaves NSPs to identify their own tactics. For example, female NSPs in Papua New Guinea asked their husbands or male colleagues to accompany them to their clients' homes during evening visits (Razee et al., 2012). In 2016, Somvati Tyagi, an ASHA in India, was raped and subsequently died by suicide during the course of her work (Dasgupta et al., 2017; Steege et al., 2018). This and other similar tragic incidents (Steege et al., 2018) have led to increased calls to formalize NSPs as health workers, giving them greater protection and access to resources, and to hold employers accountable for women worker's safety (Closser et al., 2023; Dasgupta et al., 2017). Global health programs are often "delivered by women, led by men" (Gronholm et al., 2023; WHO, 2021b) and the experiences and threats of violence, harassment and discrimination have also come from employers, upper management and co-workers (Closser et al., 2023; Mumtaz et al., 2003; Steege et al., 2018). Specific recommendations for promoting the safety of NSPs include considering the deeply patriarchal system in which NSPs often work (Dasgupta et al., 2017; Gurung et al., 2021), conducting regular anti-harassment trainings for staff to assist in organizational culture change, collecting accounts of harassment regularly during supervision or other routine meetings (Closser et al., 2023) and providing treatment, assistance and social security for NSPs affected by violence (Mishra, 2016). Additional research is needed on the incidence and impact of harassment and GBV on NSPs and the effectiveness of strategies to address these issues.

Supervision was highlighted as a significant opportunity to directly impact NSPs' experiences and personal outcomes and findings point to a need for more systematic supervision approaches. Task-sharing initiatives in LMICs currently utilize a range of approaches including traditional supervision, supportive supervision (Kemp et al., 2019), apprenticeship model (Murray et al., 2011; Rahman et al., 2019) and peer supervision (Singla et al., 2020). Studies in the scoping review recommended a range of supervision methods suggesting a holistic model with a mix of approaches that enhance technical support and personal development. Aside from a few exceptions (IFRC, 2023), there are limited in-depth descriptions or guidance on how to deliver these supervision models in low-resource settings (IFRC, 2023; Kemp et al., 2019). The Ensuring Quality in Psychological Support platform provides resources for using role-plays to assess NSPs' competencies and their readiness to deliver task-sharing interventions (Kohrt et al., 2020), especially during the training process (Watts et al., 2021). Future iterations of the platform could include additional resources for maintaining skills, motivation and personal development throughout program delivery as well as tools to measure the experiences of competency-based training and supervision from the NSPs' perspective (Vallières et al., 2018). Additional research is also needed in LMICs on how various supervision approaches impact client outcomes and NSPs' skills, confidence and overall satisfaction. Additional opportunities for programming and research include peer delivery of the mental health interventions (i.e. fully trained NSPs delivering the mental health intervention to NSP trainees as a part of the training process) and the co-creation of training and supervision methodologies with NSPs, as a part of increasing the sustainability of task-sharing interventions (Sartor & Hussian, 2024).

Organizational and health systems strengthening is also necessary to promote positive outcomes for NSPs. The need for a clear career ladder, certifications and job stability for NSPs was echoed across numerous studies in the scoping review and aligns with WHO's recommendations (WHO, 2020a). Mental health workers in Ethiopia viewed burnout as unavoidable if they continued to work without career progression and structural changes, such as adequate salary, benefits packages and supportive peers and supervisors (Selamu et al., 2017). An active referral system to existing social services, government programs and health facilities was recommended by the studies included in the scoping review. In order to develop a strong referral system and create a pathway toward future employment for NSPs, close coordination and advocacy with local agencies and governing bodies is necessary (Wainberg et al., 2017) though it is challenging in practice (Chau et al., 2021). Additional interdisciplinary research is necessary within LMICs and HICs to understand the policies and legislations related to mental health paraprofessionals. Therefore, ensuring positive outcomes for NSPs is not only beneficial at the individual level but can also support the scale-up of task-sharing interventions.

Due to the lack of literature published on this topic, the scoping review includes a limited number of publications and findings are not generalizable to all contexts. Though this was a global review, the scoping review only found relevant publications with data from eight countries. Because only two of the 15 studies were conducted in HICs, definitive comparisons between outcomes from HICs and LMICs cannot be conducted. We recommend NSP outcomes data to be collected across task-sharing studies in various contexts. The studies included in the scoping review measured outcomes of NSPs in the short term (i.e. duration of intervention delivery) and additional research is needed on evaluating the long-term impacts of working as a NSP. Longitudinal data, even after intervention delivery, should be collected on mental health impacts of working as a NSP, burnout, changes in social network, career trajectories and maintenance and utilization of therapeutic skills and competencies with their families and in subsequent community and service-related work. Much of the existing data on the impact of task-sharing interventions on NSPs have been collected through qualitative approaches and additional quantitative, mixed-methods and participatory methodologies are recommended for future research.

## Conclusion

This scoping review found that there were few studies that measured and collected data on the impacts of task-sharing interventions on NSPs. Included studies highlighted that NSPs were positively impacted by and utilized the mental health skills and competencies they learned from mental health interventions to reduce their own and their family and community's levels of distress. NSPs also faced challenges in their roles, such as harassment, difficult workplace environments, burnout and decreased time for familial responsibilities. Programming and health systems

recommendations for addressing these challenges included provision of a competitive salary and benefits, pathways toward career progression and certification, strengthened training and supervision and increased integration of NSPs' work into existing health structures. As mental health interventions are increasingly utilized in LMICs and HICs (Lange, 2021), data on the impacts on NSPs needs to be collected and considered alongside participant outcomes. Future directions for research include assessing various training and supervision and organizational/programmatic strategies and their impact on NSP outcomes. We also recommend for more task-sharing intervention studies to collect qualitative and qualitative data on the impacts of task-sharing programs on NSPs involved in delivering the mental health interventions.

**Open peer review.** To view the open peer review materials for this article, please visit http://doi.org/10.1017/gmh.2024.129.

**Supplementary material.** The supplementary material for this article can be found at http://doi.org/10.1017/gmh.2024.129.

**Data availability statement.** Authors confirm that the data supporting the findings of this study are available upon request.

**Author contribution.** MS designed the study and search strategy. JC conducted the search. JC and MS conducted the data extraction and analysis. CS developed the figures. MS drafted the manuscript and it was reviewed and edited by all authors prior to submission.

**Financial support.** This work was supported in part by funding from the US National Institute for Mental Health (grant# R01MH127767). The funders played no role in the design and conduct of the study; collection, management, analysis and interpretation of the data; preparation, review or approval of the manuscript and decision to submit the manuscript for publication.

**Competing interest.** The authors declare none.

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
