## [Reviewer Report]

This review synthesizes the evidence on the impacts of task-sharing on lay providers. In light of the great promise of task-sharing to expand access to care as well as some of the literature on burnout/task-dumping that may undermine the positive impacts of task-sharing and harm lay providers themselves, this represents a critically important topic for the field to understand better. I have made some recommendations to further refine the review below:

- Impact statement

- This reads as very similar to the abstract. I wonder if this could be written more succinctly or in more laymens/general audience terms?

- Abstract:

- Could you include the denominator here? This underscores the point that more researchers should be including this outcome.

Introduction

- “The review also aims to summarize existing recommendations at the programmatic and organizational levels to increase the positive impact of these interventions on lay providers.” Based on your content was one of the goals not also to identify recommendations that can mitigate negative impacts on these providers? I think that is a worthwhile thing to emphasize and would state it that way that too.

Methods:

- Could you include your exact search terms in supplemental materials?

- Can you say more about what you mean by “we continued our search until we reached saturation?”

- What dates did your search cover?

- I agree this makes sense to have done so but can you briefly justify why you included HIC studies too.

Results

- I find this confusingly worded: “After excluding studies that did not meet the inclusion criteria, 71 were screened by abstract and full text for relevant keywords and topics.” Can you specify more clearly how they were excluded? Was it first on title and then on abstract?

- “The backgrounds of nonspecialists that delivered the interventions were diverse.” Can you be a bit more descriptive here?

- The titles of figures are misaligned - I initially read Figure 2 as Figure 3. And Figure 3 is not mentioned in text at all. I also wonder if you could actually combine Figure 2 and 3 so the recommendations are more clearly linked to the challenges? That would help identify if there are challenges we don’t currently have recommendations for, which seems important to know. I would then also highlight learning more about challenges we don’t currently have recommendations for in the text as an area for future work.

- Were there any differences in findings across HIC and LMIC studies?

Discussion

- Love your focus on gender/safety and labor protections on page 14! This is such an important point for the field to be talking about more.

- I think this is implied, but a key point for the future directions is that more studies should be evaluating these outcomes in general.

---

## [Reviewer Report]

The authors present the results of a scoping review of the impact of task shifting on lay providers. I believe this is an important and interesting topic, as the impact of providing services on providers is generally understudied. I have offered some suggestions for the authors to consider, mainly focused on clarifying the approach and results.

Introduction:

• I think it would be helpful to provide, perhaps even in quotations, a formal definition of non-specialist providers (or whatever other term you decide to use consistently throughout the paper). Regarding the current operationalization, I do not believe all projects use non-specialists that live in the communities they service. Some choose not to due to community-level stigma.

• Related, I noted throughout the introduction and in the results that several similar terms were used throughout the manuscript (e.g., non-specialists and lay providers, project-specific terms in the results).

• The claims in the following sentence would benefit from citation: “Non-specialists, more broadly referred to as lay providers, are often trusted figures within the community and described as being uniquely positioned to reach underserved populations and even reduce the stigma associated with accessing mental healthcare.”

• The claim that task-sharing interventions are “brief scalable mental health interventions designed to be delivered by non-specialists” seems to confuse or conflate task-sharing, which is an implementation strategy, with the interventions that are being delivered via task-shifting (some of which were originally designed to be delivered by credentialed providers). I would suggest the authors review and re-write portions of the introduction to clearly distinguish between task-sharing as a strategy and not an intervention itself.

• In the sentence, “Research has revealed challenges faced by lay providers, including motivation, self-efficacy, stress, and burnout…” more information is needed on the claims introduced. Motivation and self-efficacy alone are not challenges, so further information clarifying the challenges is needed.

Methods:

• A reporting checklist, such as the PRISMA Extension for Scoping Reviews (PRISMA-ScR), would be helpful to ensure all needed information is presented.

• Regarding the search strategy, did you work with a librarian to develop the search? How was saturation determined? Did the search limit to articles published after 2010, or is that when the earliest included article was published? If the latter, I’d recommend removing the date references from the abstract as it can be somewhat confusing to readers who may interpret it as a search date.

• Was double-review completed at all stages of the screening process (i.e., title and abstract, full text, and extraction)?

Results:

• Overall, the results read as somewhat overwhelming due to the presentation of results from specific studies. I would encourage the authors to think about how to truly “scope” the literature by further synthesizing results and making higher-level claims about the current state of the science.

• A related point to the above is that by introducing the specific terminology that is used for each non-specialist provider in the respective publications (e.g., LMHWs, ASHAs), it makes it somewhat confusing to follow as a reader. I would recommend using consistent terminology throughout.

• Though the authors make specific claims in the results, they seem to cite broader papers to support their statements. For example, “However, a helper in Switzerland noted that though they could relate to their clients’ sadness, over time they were able to be empathetic (Giusto et al. 2021; Munodawafa et al. 2017) and “more resilient and less likely to be negatively affected by people’s stories” (Spaaij et al. 2023).” This sentence makes it seem as if all citations are related to helpers in Switzerland, but the Giusto article is from Kenya and Munodawafa is from South Africa.

Discussion:

• I was somewhat surprised to see the result regarding gender-based violence discussion so prominently. While I wholeheartedly agree it is a major consideration for nonspecialists, it did not seem to be presented as such in the results (I believe there was only one sentence with two citations that mentioned this in the results). As such, and to highlight this point without perhaps unintentionally overstating the results of your scoping review, I might suggest to explicitly name the limited research on gender-based violence against non-specialists as a finding itself.

• With more scoping of the current literature, I wonder if the authors might also be able to make some recommendations on future directions for research in this area. Currently, much of the discussion is recapping findings, and I think it would be useful to spend more time discussing the implications of this review for future work.

---

## [Reviewer Report]

Thank you for the opportunity to review this great article that sheds more light on the positive and negative impacts on non-specialists as they provide task-sharing interventions along with important recommendations.

Abstract:

In noting “across 8 countries” it could be helpful to offer more context that the countries represented 3-4 global regions and were primarily LMICs with a few HICs.

Impact Statement:

In the middle of the paragraph, “A total of 15 studies that collected quantitative or qualitative data…” could be edited to read “and/or” given further clarification in the article that some articles had both quantitative or qualitative data.

In the last sentence, “non-specialists’ outcomes” at first read could leave a reader thinking this refers to the outcomes from the intervention delivered by non-specialists on participants. I am flagging this possible confusion for consideration from the authors, this wording appears a few times in the article in other sections as well. Instead, the “impact on non-specialists” language could be used consistently.

Introduction:

I was struck by the similarities in the findings and recommendations in this article focused on mental health service delivery on non-specialists to that of the impact on mental health service delivery on mental health specialists. Have the authors considered noting any similarities or differences across these two groups? I think acknowledgement of existing literature focused on mental health specialists could help a reader place this information in more context and demonstrate why this article, and further research, is needed as noted in the article.

Individual-family-and community-level impacts on non-specialists:

In the first paragraph, third sentence, I think “cope” should be in the past tense to match the tense of the sentence.

Facilitators, Barriers, and Recommendations for increasing positive outcomes for non-specialists:

I think it could be helpful to add a note in the narrative referencing Figure 3.

Starting on page 11, Line 31- it is possible to describe the frequency or prevalence of the findings? For example, adding a term like consistently overwhelmingly, often, sometimes, occasionally, etc. in each of the first few sentences that do not have a specifying term included. It was challenging to have a grasp of how clear these findings came through. Or another way to help a reader understand could be to rank the prevalence across articles of the themes identified, or offer a percentage of how often the recommendation was raised across the articles.

I was curious if the Facilitators, Barriers, and Recommendations varied between regions and/or between HICs and LMICs. If there were any findings to share here this could be of interest to reader.

---

## [Editor Report]

Dear Authors:

Thank you for the opportunity to review this manuscript on the impact of task-sharing mental health interventions on non-specialists. 

The Reviewers provide substantial feedback that I hope you may consider when revising your manuscript. 

I’d ask you to especially consider these points (some of which were also raised by the reviewers) during your revision:

1. Differentiate better between the Impact Statement and the Abstract. This is an opportunity to accentuate *why* this study is so important (Impact Statement) vs. a summary of what you did (abstract).

2. Greater methodological description including the use of a PRISMA checklist for scoping reviews.

3. Refine how lay workers were defined and use the selected term consistently throughout the manuscript. In particular, de-link the role of the lay worker from the intervention type (e.g., not all lay workers deliver brief interventions). Further, recommend dropping the licensure differentiation between lay workers and specialists (unless that can be verified for all the studies in the review.

3. Check to ensure use of non-stigmatising language: “seropositive” on page 10 line 4: consider “living with HIV” (preferred by IAS); “committed suicide” on page 14 line 8: consider “died by suicide” (preferred by WHO).

---

## [Reviewer Report]

Thanks again for the opportunity to review. I also appreciated reviewing the comprehensive revisions that further strengthened this article.

---

## [Reviewer Report]

Thank you for your close attention to my comments. I have only one minor edit to add.

The first sentence the abstract “Non-specialist providers (NSPs), people without specialized clinical training, are increasingly utilized to deliver task-sharing approaches, such as scalable mental health interventions” is slightly confusing, in my opinion, as NSPs are the only people to deliver task-sharing approaches (and therefore not increasingly utilized to deliver them but have always delivered them). Rather, I think it more clear that “Task-shifting is increasingly used to address the global mental health treatment gap.”

---

## [Editor Report]

Dear Authors - thanks you for this revised version of your manuscript.

Two minor issues are pending:

1. Licensure - we note that “licensure” and “license” were removed from the manuscript body but retained in the definition of a NSP in the impact statement. In the first review, a recommendation was "Further, recommend dropping the licensure differentiation between lay workers and specialists (unless that can be verified for all the studies in the review). Can you verify that the NSPs in the articles reviewed made this distinction?

3. Please see the minor pending comment from Reviewer 3.

Many thanks for your attention to these issues.

---

## [Reviewer Report]

Unfortunately, I think the new edit conflates task-sharing (the implementation strategy) with the intervention (scalable mental health interventions). Scalable mental health interventions are not task sharing approaches--they are just interventions. Likewise, task-sharing approaches are just that--approaches to intervention delivery. A task sharing approach can be used to deliver any intervention. A “scalable” mental health intervention can also be delivered by a credentialed provider.

---

## [Editor Report]

Dear Authors - please see this Reviewer’s comment regarding task-sharing (an approach) and scalable interventions; these are separate concepts (the former “how to” and the latter “what is done”). Some interventions use task-sharing, while others do not. Could you please addresses this issue more clearly?